# MicroRNAs in Diffuse Large B-Cell Lymphoma (DLBCL): Biomarkers with Prognostic Potential

**DOI:** 10.3390/cancers17081300

**Published:** 2025-04-12

**Authors:** Yuliya A. Veryaskina, Sergei E. Titov, Igor B. Kovynev, Sofya S. Fyodorova, Olga V. Berezina, Igor P. Zhurakovskij, Oksana V. Antonenko, Sergei A. Demakov, Pavel S. Demenkov, Pavel S. Ruzankin, Anton S. Tarasenko, Tatiana I. Pospelova, Igor F. Zhimulev

**Affiliations:** 1Laboratory of Molecular Genetics, Department of the Structure and Function of Chromosomes, Institute of Molecular and Cellular Biology, SB RAS, Novosibirsk 630090, Russia; titovse78@gmail.com (S.E.T.); ovant@mcb.nsc.ru (O.V.A.); demakov@mcb.nsc.ru (S.A.D.); zhimulev@mcb.nsc.ru (I.F.Z.); 2Laboratory of Gene Engineering, Institute of Cytology and Genetics, SB RAS, Novosibirsk 630090, Russia; 3AO Vector-Best, Novosibirsk 630117, Russia; 4Department of Therapy, Hematology and Transfusiology, Novosibirsk State Medical University, Novosibirsk 630091, Russia; kovin_gem@mail.ru (I.B.K.); soffka.tv@mail.ru (S.S.F.); ovb-mail@ya.ru (O.V.B.); murash2003@yandex.ru (I.P.Z.); depart04@mail.ru (T.I.P.); 5Laboratory of Computer Proteomics, Institute of Cytology and Genetics, SB RAS, Novosibirsk 630090, Russia; demps@bionet.nsc.ru; 6Department of Probability Theory and Mathematical Statistics, Novosibirsk State University, Novosibirsk 630090, Russia; ruzankin@math.nsc.ru (P.S.R.); a.tarasenko@g.nsu.ru (A.S.T.); 7Sobolev Institute of Mathematics, Novosibirsk 630090, Russia

**Keywords:** microRNA, diffuse large B-cell lymphoma, biomarkers

## Abstract

Diffuse large B-cell lymphoma (DLBCL) is a heterogeneous disease from both genetic and clinical points of view. Thus, new biomarkers for the diagnosis and prognosis of DLBCL, as well as deciphering the mechanism of action of deregulated microRNAs in the origin of the disease, are needed. From a practical point of view, the results obtained in this study may contribute to the development of personalized therapy. From a fundamental point of view, the obtained data will contribute to the understanding of the pathogenesis of DLBCL.

## 1. Introduction

Diffuse large B-cell lymphoma (DLBCL) is the most common type of non-Hodgkin lymphoma. From a clinical perspective, DLBCL refers to a heterogeneous group of B-cell lymphomas. A significant percentage of DLBCL patients achieve effective remission after therapy. However, relapse eventually occurs in approximately 40% of patients [1]. The International Diagnostic Index (IPI) Scoring System is the most widely used prognostic grading system and involves the following five factors: age, the ECOG performance status, lactate dehydrogenase (LDH) level, clinical stage and number of affected extranodal organs but does not allow for the molecular characteristics of a tumor. Since DLBCL is an extremely heterogeneous tumor, identical clinical pathological characteristics of two patients do not necessarily mean that these patients have the same prognosis. Today, at least four prognostic systems need to be compared: IPI, R-IPI (with allowance for adding rituximab (R) to the therapy regimen), the National Comprehensive Cancer Network-IPI (NCCN-IPI) scoring system, and the Grupo Español de Linfomas y Trasplantes de Médula Ósea (GELTAMO) (G-IPI) scoring system [2].

Recent studies have demonstrated that risk models based on gene expression can improve the prognostic accuracy for DLBCL. He et al. developed a DLBCL prognostic panel consisting of 14 metabolism-associated biomarkers (*NR3C1, IGFBP3, RARRES2, F5, APOC1, CSF2RA, ENPP1, GYG1, PHKA1, CPT1A, PDK4, CLOCK, CTH,* and *PLTP*) [3]. Cui et al. showed a prognostic model for DLBCL based on the analysis of the *PD-L2, CTLA4, TIM-3, TIGIT* and *B7-H3* glycolysis-related genes [4]. Chen et al. developed a prediction panel consisting of eight ferroptosis-related markers (*ZEB1, PSAT1, NGB, NFE2L2, LAMP2, HIF1A, FH,* and *CXCL2*) [5]. Based on the analysis of 12 immunogenic cell death-related genes, Ping et al. developed a DLBCL prediction panel [6]. Zhang et al. proposed the DLBCL risk rating model based on analyzing 16 lipid metabolism-related markers (*ACSM3, ARSK, CEPT1, DGKE, EHHADH, ENPP7, FABP4, FASN, LPGAT1, MECR, PTDSS2, RAN, ARSJ, CYP27A1, FAM120B,* and *PIK3CG*), which can compensate for the limitations of the IPI scoring system [7]. Pan et al. reported that the high expression levels of five genes (*CEBPA, CYP27A1, LST1, MREG,* and *TARP*) are associated with favorable prognosis in DLBCL patients [8]. Carreras et al. demonstrated that the levels of *ARG1, TNFSF12, REL, NRP1, IFNA8, CASP1,* and *CTSG* correlated with estimated survival in DLBCL [9]. In their study, Chen et al. demonstrated that *CD3G, CD3D, GNB4, FCHO2,* and *GPR183* are involved in regulation of the immune microenvironment in DLBCL and can be used as biomarkers for predicting immunochemotherapy response as well as for potential therapeutic applications in DLBCL patients [10]. Wright et al. created the LymphGen algorithm, which allows one to classify DLBCL into genetic subtypes by means of genetic aberrations. This classification subdivides DLBCL into seven genetic subtypes that differ in oncogenic-pathway engagement, patients’ survival rates, the gene expression phenotype, tumor microenvironment, and possible therapeutic targets [11].

Analysis of microRNA (miRNA) expression is a novel tool for understanding the biology and the clinical course of DLBCL. miRNAs are ~22 nt long endogenous small noncoding RNAs. miRNAs were shown to be involved in many cellular processes, and their aberrant expression facilitates the development of different types of tumors [12]. Hu et al. demonstrated that analysis of miRNA-296-5p and miRNA-1304-5p expression, the IPI status, and patient age were independent indicators of DLBCL prognosis [13]. Another study showed that shorter overall survival was observed for DLBCL patients in the case of low miRNA-129-5p expression, thus characterizing this miRNA as an unfavorable prognostic marker [14]. Shepshelovich et al. demonstrated that the expression profiles of miRNA-17-5p, -19b-3p, -20a-5p, -106a-5p, -150-5p, -342-3p, -181a-5p, and -140-3p make it possible to differentiate between DLBCL patients having a favorable and an unfavorable prognosis [15].

The IPI system continues to remain the most reliable prognostic tool for DLBCL patients and is used in all clinical trials. However, the heterogeneity of DLBCL based on differences in both genetic and epigenetic factors undoubtedly contributes to the dynamics of tumor growth. Research into a large number of molecular biomarkers involved in the pathogenesis of DLBCL will enable the elaboration of a more perfect DLBCL prognostic and diagnostic system and treatment personalization.

The purpose of this study was to identify biomarkers that can improve the accuracy of DLBCL diagnosis and of prognosis estimation.

## 2. Materials and Methods

### 2.1. Clinical Samples

Forty-three formalin-fixed paraffin-embedded (FFPE) lymph node (LN) samples from primary DLBCLs were included in the study.

One-hundred and five cytological samples were collected via sternal puncture and bone marrow (BM) aspiration biopsy from the posterior iliac spine. The analyzed groups included DLBCL (*n* = 70) and a control group, composed of patients with non-cancerous blood disorders (NCBDs) (*n* = 35) [hemolytic anemia (*n* = 5), iron deficiency anemia (*n* = 20), immune thrombocytopenia (*n* = 5), and B12 deficiency anemia (*n* = 5)]. The tissue samples under study were obtained from the State Novosibirsk Regional Clinical Hospital and City Clinical Hospital No. 2 (Novosibirsk).

Appendix A lists the groups’ characteristics. The tissue samples were collected in conformance to laws and regulations of Russia; all patients provided written informed con-sent, and all their data were anonymized. Tissue sampling for this work was initiated on 10 January 2022, and completed on 1 May 2024. The study was carried out in accordance with the Declaration of Helsinki and was approved by the Ethics Committee at Novosibirsk State Medical University (study protocol No. 15; 25 May 2020).

### 2.2. Isolation of Total RNA from FFPE Samples

For the deparaffinization, 1 mL of mineral oil was put into a microcentrifuge tube with three 15 μm paraffin-embedded slices of lymph node tissue. Then, the microfuge tube was vortexed for 10 s and held in a thermal shaker at 1300 rpm and at 65 °C (BioSan, Riga, Latvia) for 2 min. After that, the samples were centrifuged at 13,000–15,000× *g* for 4 min, and the supernatant was thrown away without disturbance of the precipitate. Ethanol (96%, 1 mL) was added to the pellet, with subsequent vortexing for 10 s and centrifugation for 4 min at 13,000–15,000× *g*. The supernatant was thrown away without disruption of the sediment, with subsequent addition of 70% ethanol (one mL) and centrifugation for 2 min at 13,000–15,000× *g*. The resultant pellet was subjected to isolation of nucleic acids.

Guanidine lysis buffer (600 µL) was mixed with each sample. Next, the samples were mixed thoroughly and incubated in a thermal shaker for 15 min at 65 °C (BioSan, Riga, Latvia). After that, the same volume of isopropanol was introduced. The mixture was flipped energetically a few times and kept at room temperature for 5 min. After centrifugation at 14,000× *g* for 10 min, the supernatant was cautiously decanted, and the sediment was washed by means of 70% ethanol (500 μL) and then acetone (300 μL). Finally, the prepared RNA was dissolved in deionized water (290 microliters).

### 2.3. Isolation of Total RNA from Fine-Needle Aspiration Cytological Samples

Every dried cytological sample was washed with portions (200 µL × 3 times) of guanidine lysis buffer in a microfuge tube. The samples were thoroughly mixed and kept at 65 °C for 15 min in a thermal shaker. After that, isopropanol was introduced (equal volume). Next, the resultant solution was vigorously mixed and incubated for 5 min at room temperature. After centrifugation for 10 min at 14,000× *g*, the supernatant was carefully removed, and the precipitate was washed with 70% ethanol (500 µL) and acetone (300 µL). The obtained RNA sample was dissolved in deionized water (290 µL).

### 2.4. RNA Sequencing

RNA was employed to generate libraries for massively parallel sequencing by means of the NEBNext Small RNA Library Prep Set for Illumina E7330S (New England Biolabs, Ipswich, MA, USA) according to instructions in the kit. Library concentration was measured with the Qubit dsDNA HS Assay Kit (Thermo Fisher Scientific (Waltham, MA, USA)) with the help of a Qubit 2.0 fluorometer. The distribution of fragment lengths was determined with the Agilent High Sensitivity DNA Kit (Agilent Technologies, Santa Clara, CA, USA). Sequencing was conducted by means of HiSeq1500 (Illumina, CA, USA) with the obtaining of at least 5 × 10^6^ short (50-nucleotide) sequencing reads.

Adapters’ sequences were eliminated from the reads with the help of cutadapt v3.1; the sequencing reads were then mapped onto the human genome (GRCh38) in bowtie version 1.0.1. Quantification of microRNA in the resultant alignments was conducted by means of a feature in Counts version 1.5.0-p1 software along with minOverlap 14-Q 10 parameters for database mirbase v22.

### 2.5. Validation of RNA Sequencing Results by RT-qPCR Analysis

The sequences of all oligos are listed in Appendix A. The geometric mean of miR-378, miR-103a and miR-191 expression levels was used for normalization. Oligonucleotides were selected using the PrimerQuest online service (http://eu.idtdna.com/ (accessed on 10 January 2018)). The qPCR threshold cycles were analyzed using the 2^−ΔCt^ method.

### 2.6. Reverse Transcription

Reverse transcription of cDNA was performed in a reaction mixture of 30 µL. The Ready-to-use RealBest RT Master Mix (Vector-Best, Novosibirsk, Russia) was employed. The reaction contained 3 μL of an RNA specimen, 0.4 mM each dNTP, 21.6% of trehalose, 1xRT buffer (Vector-Best, Novosibirsk, Russia), 1% of BSA, 100U of the M-MLV reverse transcriptase (Vector-Best, Novosibirsk, Russia), and 0.2 μM appropriate RT primer. All the oligos were synthesized at Vector-Best (Vector-Best, Novosibirsk, Russia). The reaction mixture (3 µL) containing cDNA was subjected immediately as a template to real-time PCR by means of a CFX96 (Bio-Rad, Hercules, CA, USA) system.

### 2.7. Real-Time PCR

The miRNA expression levels were measured by real-time PCR on a CFX96 amplifier (Bio-Rad Laboratories, Hercules, CA, USA). The total volume of each reaction was 30 μL; the reaction mixture contained 3 μL of cDNA, 1× PCR buffer (Vector-Best, Novosibirsk, Russia), 0.4 mM of each dNTP (Biosan, Riga, Latvia), 1% BSA, 1U Taq polymerase (Vector-Best, Novosibirsk, Russia) premixed with 10× active site-specific monoclonal antibody (Clontech, Mountain View, CA, USA), 0.5 units of uracil-DNA glycosylase (Vector-Best, Novosibirsk, Russia), 0.5 μM of each primer, and 0.25 μM of TaqMan probe. The primers and the probes had been developed by Vector-Best (Vector-Best, Novosibirsk, Russia); PCR efficiency was 90–100%.

### 2.8. Reverse-Transcription PCR

Real-time reverse-transcription PCR (RT-PCR) was carried out with specific primers and fluorescently labeled probes to detect an mRNA and a housekeeping gene’s mRNA (*PGK1;* phosphoglycerate kinase) for normalization. The miRNA expression levels were measured by real-time PCR on a CFX96 amplifier (Bio-Rad Laboratories, Hercules, CA, USA). The total volume of each reaction was 30 μL. The reaction mix contained 3 μL of RNA preparation, 21.6% trehalose, 1xRT-PCR buffer (Vector-Best, Novosibirsk, Russia), 0.4 mM of each dNTP (Biosan), 1% BSA, 100U M-MLV reverse transcriptase (Vector-Best, Novosibirsk, Russia), 1U Taq polymerase (Vector-Best, Novosibirsk, Russia) pre-mixed with 10× active center-specific monoclonal antibody (Clontech, Mountain View, CA, USA), 0.5 μM of each primer and 0.25 μM of Taqman probe.

### 2.9. Statistical Analysis

This analysis was conducted with the help of STATISTICA v13.1 software. The Mann–Whitney *U* test was chosen; results with a *p* value < 0.05 were regarded as statistically significant.

Bioinformatic analyses of target genes of miRNAs were executed with the help of the miRNet 2.0 tool (https://www.mirnet.ca/miRNet/home.xhtml; accessed on 1 October 2024).

## 3. Results

### 3.1. MicroRNAs as Diagnostic Biomarkers in Bone Marrow Samples in Patients with DLBCL and Non-Cancerous Blood Diseases

MiRNA sequencing analysis was performed on seven bone marrow (BM) samples from patients with DLBCL and six BM samples from patients with NCBD (Figure 1). The criteria for inclusion of an miRNA in subsequent analyses were as follows: a median number of reads greater than 1000 for at least one group and a difference in the amount of the miRNA greater than twofold between groups being compared. On the basis of these conditions, a list was compiled of 62 miRNAs that are differentially expressed between DLBCL and NCBD in BM samples (Appendix A).

Next, we conducted a comparative analysis of the expression levels of miRNA-182, -451a, -151a, -148b, -183, -192, and miRNA-106b in bone marrow samples from DLBCL patients with bone marrow involvement (*n* = 18), DLBCL patients without bone marrow involvement (*n* = 52), and patients with non-cancerous blood diseases (NCBDs) (*n* = 35). These miRNAs were chosen because they showed statistically significant differences in the comparison of sequencing results between DLBCL and NCBD (Appendix A) and are known to be associated with DLBCL [16,17,18,19]. Statistically significantly increased expression levels of miRNA-151a, miRNA-148b and miRNA-192 were observed in bone marrow samples from DLBCL patients both with and without bone marrow involvement compared to bone marrow samples from NCBD patients (*p* < 0.05) (Figure 2). Therefore, it can be inferred that expression analysis of miRNA-151a, miRNA-148b and miRNA-192 in bone marrow samples is a potential diagnostic approach for detecting localized DLBCL.

### 3.2. Diagnostic Genetic Biomarkers in BM Samples of Patients with DLBCL or Non-Cancerous Blood Diseases

Based on the literature data, we selected 11 putative biomarkers for DLBCL: *ASF1B, CD82B, CRISP3, FN1, MEF2B, PD-L1, TIMP1, TOP2A,* and *TP53*. These genes are oncogenes and suppressor genes of different tumor types [20,21,22,23,24,25,26,27,28]. We wished to determine whether they show anomalous expression in DLBCL as well. The presence of variation of their expression in DLBCL suggests that they are possible biomarkers of DLBCL.

It was found that the *PD-L1* mRNA level is elevated both in DLBCL without BM involvement and in DLBCL with BM involvement in comparison with NCBD (*p* < 0.05); mRNA levels of *TIMP1*, *TOP2A*, and *TP53* proved to be lowered both in DLBCL without BM involvement and in DLBCL with BM involvement as compared to NCBD (*p* < 0.05) (Table 1).

### 3.3. MiRNAs as Prognostic Biomarkers of DLBCL in Lymph Nodes and Their Impact on Survival

Retrospectively, we compiled a group of patients with DLBCL who had received a standard cytoreductive therapy and had completed a remission-inducing program. The study included patients that responded to the treatment with either recurrence-free remission (“favorable prognosis”) or with progressive disease without remission (“poor prognosis”). On the basis of literature data, we selected miRNA-100, -125a, -125b, -126a, -143, -155, -16, -197, -221, -23a, -26a, -30b, -486, -574, -7, -10b, and -146a and let7a, which are characterized by aberrant expression in B-cell lymphomas [29,30]. By real-time RT-PCR, we performed comparative analysis of the miRNA expression levels between DLBCL groups with a poor prognosis (*n* = 36) and a favorable prognosis (*n* = 7). We observed statistically significant reduction in the expression levels of miRNA-10b, -100, -125a, -125b, -126, -143, -23a, and let-7a in the group of DLBCL patients having a poor prognosis vs. DLBCL patients having a favorable prognosis (*p* < 0.05) (Figure 3).

The overall survival (OS) curves were plotted for patients; the patients were divided into high- and low-expression groups in accordance with the mean expression of each out of eight miRNAs statistically significantly related to prognosis (Figure 4). Kaplan–Meier survival analysis demonstrated that the high expression levels of miRNA-125a (*p* = 0.03), miRNA-23a (*p* = 0.05), and miRNA-100 (*p* = 0.05) in DLBCL patients were associated with favorable prognosis and better overall survival.

### 3.4. Analysis of the Network of Interactions of mRNAs and miRNAs in Lymph Nodes

Next, we conducted bioinformatics analysis of the target genes encoding miRNAs -10b, -100, -125a, -125b, -126, -143, -23a and let-7a, which are statistically significantly associated with prognosis in DLBCL. From the Kyoto Encyclopedia of Genes and Genomes (KEGG) database, we obtained data on the miRNA target genes involved in such biological pathways as “Pathways in cancer”, “Cell cycle”, “Apoptosis”, and “B-cell receptor signaling pathway”. We identified a total of 90 targets using the MiRNet database. Some target genes are simultaneously regulated by several miRNAs. Thus, *AKT1* is regulated by miRNA-10b, -100, -125a, -125b, -126, -143; *BCL2* is regulated by miRNA-125a, -125b, -126, -143 and let-7a; *MYC* is regulated by miRNA-125a, -23a and let-7a; *PIK3R1* is regulated by miRNA-126, -143, -23a and let-7a; *IGF1R* is regulated by miRNA-100, -125b, -143 and let-7a; *KRAS* is regulated by miRNA-126, -143 and let-7a; *EGFR* is regulated by miRNA-125a, -125b and let-7a; *CDKN1A* is regulated by miRNA-10b, -125a and let-7a; *XIAP* is regulated by miRNA-10b, -125b, -143 and -23a; and *STAT3* is regulated by miRNA-125a, -125b, -23a and let-7a. The data obtained are visualized using the miRNet 2.0 resource (Figure 5). MiRNet is a miRNA network visual analytics platform (https://www.mirnet.ca/miRNet/home.xhtml (accessed on 1 October 2024)).

## 4. Discussion

We analyzed the expression levels of 18 miRNAs in lymph node samples in the group of DLBCL patients who had undergone highly efficacious first-line therapy courses (and therefore had a favorable prognosis) and lymph node samples from DLBCL patients with persistently relapsing cancer resistant to anticancer therapy, which were responsible for poor prognosis and low overall survival. It has been found that expression levels of miRNA-10b, -100, -125a, -125b, -126, -143, -23a and let-7a were statistically significantly reduced in the group of DLBCL patients having a poor prognosis compared to DLBCL patients with a favorable prognosis (*p* < 0.05). Kaplan–Meier survival analysis demonstrated that the increased expression levels of miRNA-23a, miRNA-125a, and miRNA-100 are associated with better overall survival in DLBCL patients.

miRNA-23a is among the best-studied miRNAs in different human cancers [31]. The miRNA-23a expression level in tumor tissues can be both upregulated and downregulated depending on the type of cells analyzed [32,33,34]. An analysis of literature data demonstrated that miRNA-23a can function both as an oncogene and as a tumor suppressor gene [35,36]. miRNA-23a modulates not only oncogenesis but also hematopoiesis. miRNA-23a regulates a complex array of transcriptional and signaling pathways for modulation of hematopoiesis [37]. Therefore, aberrant expression of miRNA-23a is undoubtedly involved in the development of hematological tumors. Thus, MiRNA-23a and miRNA-23b are required for proper production and differentiation of hematopoietic precursor cells [38]. Kurkewich et al. noted that the cluster of miRNA-23a, -24-2, and -27a is involved in regulation of immune cell populations via suppression of B lymphopoiesis [39]. Kong et al. also confirmed the role played by miRNA-23a in the regulation of B-lymphopoiesis both in vitro and in vivo [40].

Low expression of miRNA-23a is associated with the aggressive and unfavorable prognostic phenotype of prostate cancer, unfavorable prognosis for mucosal melanoma, and poor prognosis for gastric cancer [41,42,43]. The high miRNA-23 expression is associated with poor prognosis in laryngeal squamous cell carcinoma [44]. Meanwhile, miRNA-23a-3p suppresses cell proliferation in oral squamous cell carcinoma by targeting *FGF2* and correlates with a better prognosis [45]. MiRNA-23a is involved in the pathogenesis of both solid tumors and hematological malignancies. In their study, Mi et al. demonstrated that miRNA-23a, together with miRNA-24 and miRNA-27a, is differentially expressed between acute myeloid leukemia and acute lymphoblastic leukemia [46]. The published data on DBCLC are contradictory. Thus, Wang et al. showed that DLBCL patients with a high miRNA-23a expression level typically had a shorter overall survival [30]. Xu et al. demonstrated that miRNA-23a promoted proliferation, invasion and metastatic invasion of DLBCL cells through inhibition of *MTSS1* [47]. However, the findings reported by Perry et al. showed a worse overall survival in DLBCL patients with downregulated miRNA-23a expression, consistent with our data [48].

It has been demonstrated in many studies that miRNA-100 was dysregulated in many cancer types, thus indicating that miRNA-100 can be a potential biomarker of cancer diagnosis and prognosis. Leite et al. have arrived at an interesting conclusion: they demonstrated that miRNA-100 can act both as an oncogene and as a tumor suppressor in the same tumor type depending on clinical factor being analyzed [49]. Lin et al. showed that upregulated miR-100 expression in mantle cell lymphoma cells may inhibit proliferation, promote apoptosis, and inhibit the cell cycle at G1 phase through targeting the *mTOR* gene [50]. Importantly, mantle cell lymphoma is a B-cell lymphoma, and the regulatory mechanisms of miRNA-100 in carcinogenesis will also be similar in the case of DLBCL, but further research is required.

Meta-analyses have demonstrated that low miRNA-100 expression in tumor tissues can be associated with a worser prognosis with statistical significance [51,52]. A meta-analysis of 36 studies focusing on laryngeal cancer showed that high expression of miRNA-34a, -100, -21, -155, and let-7 is associated with low overall survival [53]. A prognostic model for gastric cancer was elaborated based on analyzing expression of miRNA-100, miRNA-374a, miRNA-509-3, miRNA-668, miRNA-549, and miRNA-653 [54]. Although most publications consider a reduction in the miRNA-100 expression level a negative factor in cancer prognosis, Bai et al. demonstrated that the increased level of miRNA-100 expression correlates with a poor prognosis in children with acute myeloid leukemia [55]. Our findings agree with the data reported by Lawrie et al., who demonstrated that increased miRNA-100 expression is typical of DLBCL cases with a good prognosis [56].

Literature review demonstrated that in most studies, miRNA-125a is shown to act as a tumor suppressor [57,58,59,60,61,62]. Meanwhile, miRNA-125a was reported to have oncogenic potential, which may promote the progression and invasion of gastric cancer through the regulation of *EphA2, TAZ,* and *TEAD2* and myeloma progression through a reduction in *DIS3* expression level [63,64]. Guo et al. emphasized that miRNA-125a controls hematopoietic stem cell number [65].

From the perspective of using miRNA-125a as a prognostic biomarker, meta-analysis of ten different types of cancer demonstrated that upregulated expression of miRNA-125a-5p is a marker of favorable prognosis [66]. In our study, we have confirmed this trend, having shown that the upregulated level of miRNA-125a expression is a favorable prognostic marker in DLBCL (*p* = 0.03).

Bioinformatics analysis demonstrated that miRNA-10b, -100, -125a, -125b, -126, -143, -23a, and let-7a are targeting oncogenes and tumor suppressor genes. Hence, aberrant expression of these miRNAs contributes to tumor development and progression.

Thus, *AKT1* is simultaneously regulated by six of the miRNAs studied by us (miRNA-10b, -100, -125a, -125b, -126, -143); *MYC* is regulated by miRNA -125a, -23a, and let-7a; *IGF1R* is regulated by miRNA -100, -125b, -143, and let-7a; *XIAP* is regulated by miRNA -10b, -125b, -143, and -23a; and *STAT3* is regulated by miRNA -125a, -125b, -23a, and let-7a. Sun et al. demonstrated that *AKT1* is involved in oncogenesis; its enhanced activity was detected in primary prostate, breast, and ovarian cancer [67]. Furthermore, higher *AKT1* expression is associated with poor prognosis for endometrial cancer [68]. *MYC* triggers selective gene expression amplification to promote cell growth and proliferation [69]. IGF plays a crucial role in cancer development and metastatic spread. Type 1 IGF receptor (*IGF-1R*) is a key IGF axis component and is considered to be an oncogene in several cancer cell lines [70]. Patients diagnosed with head and neck squamous cell carcinoma (HNSCC) and having a high *IGF-1R* expression level had a significantly shortened overall survival [71]. *XIAP* (X-linked IAP) suppresses apoptosis by directly binding to caspases and inhibiting them [72]. *STAT3* is an oncogene, and its expression level is elevated in patients with breast cancer as well as head and neck squamous cell carcinoma [73].

We demonstrated that upregulated expression of miRNA-125a, miRNA-23a, and miRNA-100 correlates with a favorable prognosis in DLBCL patients. It is possible that one of the factors related to this result is targeting oncogenes (specifically, *AKT1, IGF1R, XIAP, STAT3,* and *MYC*) by these miRNAs.

We observed a statistically significant increase in the expression of miRNA-151a, miRNA-148b, and miRNA-192 in bone marrow samples from DLBCL patients both with and without bone marrow involvement compared to bone marrow samples from patients with NCBD (*p* < 0.05).

The multi-year research into functions of miRNAs demonstrated that they play a key role in normal hematopoiesis. In particular, miRNA-125b, -196a, -196b, -130a, let-7d, -148b, and miRNA-351 were found to be typically activated in stem cell and precursor populations compared to more differentiated cell types, while miRNA-151, -152, -184, -187, -212, -30a-3p, -30e-5p, and miRNA-451 are involved in the development of erythroid cells [74]. Rajasekhar et al. demonstrated that the expression levels of miRNA-192, -151, -92, and -126 differ at least twofold between myeloid precursors and respective monocytes and granulocytes [75]. Therefore, aberrant expression of these miRNAs can promote the development of hematological malignancies.

To date, miRNA expression profiling studies have revealed aberrant expression of miRNA-126 in various cancers. A meta-analysis indicates that miRNA-126 overexpression is a favorable prognostic factor for overall survival in various cancers [76]. We also noticed a trend toward miRNA-126 overexpression in the good-prognosis group, thereby confirming the role of miRNA-126 as a tumor suppressor.

Along with our study, there are currently a number of publications aiming to form miRNA panels that can differentiate DLBCL patients depending on prognosis. Larrabeiti-Etxebarria et al. demonstrated that the expression of seven miRNAs (miRNA-4444, -449c-5p,-3681-5p, -3928-3p, -449b-5p, -370-3p, and miRNA-4424) was significantly upregulated in patients with long-term remission, while the expression of three miRNAs (miRNA-133a-3p, miRNA-208b-3p, and miRNA-205-5p) was significantly upregulated in patients with recurrence; furthermore, high expression of miRNA-370-3p was associated with better five-year survival in DLBCL patients [19]. A systematic review demonstrated that the upregulated expression of miRNA-222-3p and miRNA-155-5p is associated with a worse prognosis in DLBCL patients [77]. Bento et al. noted that the expression levels of miRNA-1244, -1231, -193b-5p, -885-3p, -182-5p, and miRNA-199a-5p correlate with the overall survival in DLBCL patients [78]. Zhong et al. revealed that the expression levels of miRNA-155 and miRNA-146a in FFPE samples from DLBCL patients had prognostic significance [79]. Zheng et al. analyzed the data reported in 18 studies and arrived at a conclusion that the high expression of miRNA-21 and low expression of miRNA-224 in the tumor tissue of DLBCL patients are associated with poor overall survival [80].

## 5. Conclusions

Although searching for prognostic markers for DLBCL is a relevant contemporary problem of molecular biology, the data obtained still have not been integrated with routine clinical analysis of the assessment of prognosis for DLBCL. In the present work, we did not find statistically significant differences in the expression of the miRNAs under study depending on IPI status of the cancer. This finding may be explained by our small sample size. Therefore, our results are difficult to integrate into the existing system of prognosis. Further multicenter trials involving large sample sizes are needed for validation of our data.

## Figures and Tables

**Figure 1 cancers-17-01300-f001:**
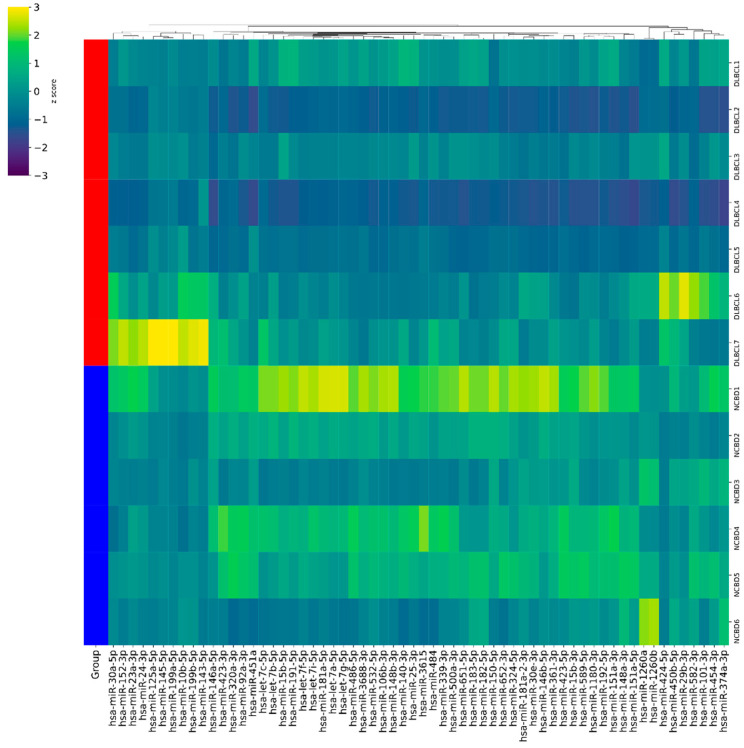
Hierarchical cluster analysis between 7 diffuse large B-cell lymphoma (DLBCL) cases and 6 cases of non-cancerous blood diseases (NCBDs) for miRNAs that were selected for validation using RT-PCR in the groups being investigated. Each column illustrates the expression of an miRNA, and each row means a nucleic acid specimen. Yellow: overexpressed miRNA; dark blue: downregulated miRNA; green: minor changes; red: a graphical representation of a group of samples.

**Figure 2 cancers-17-01300-f002:**
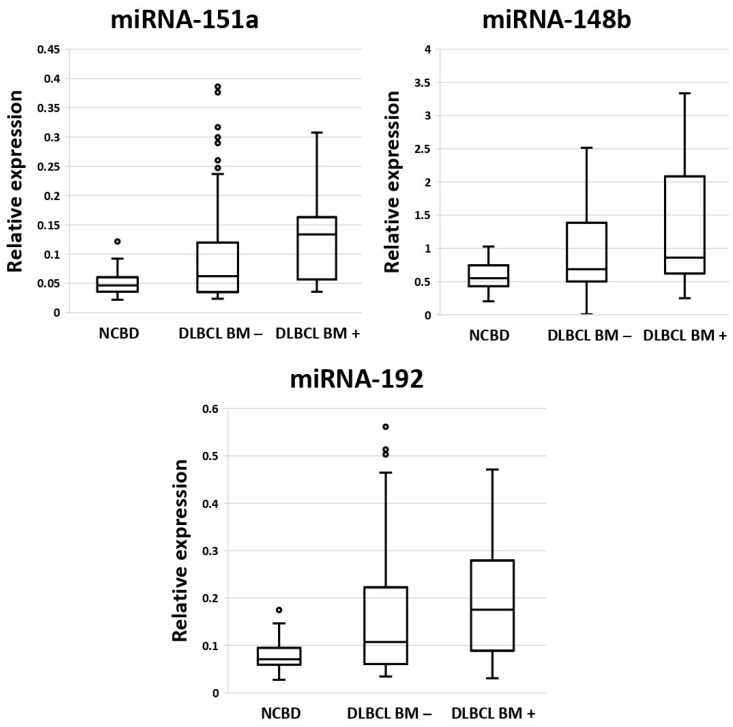
The comparative analysis of miRNA expression levels between bone marrow samples from patients with diffuse large B-cell lymphoma with bone marrow involvement (DLBCL BM+) (*n* = 18), from patients without diffuse large B-cell lymphoma without bone marrow involvement (DLBCL BM−) (*n* = 52), and from patients with non-cancerous blood diseases (NCBDs) (*n* = 35).

**Figure 3 cancers-17-01300-f003:**
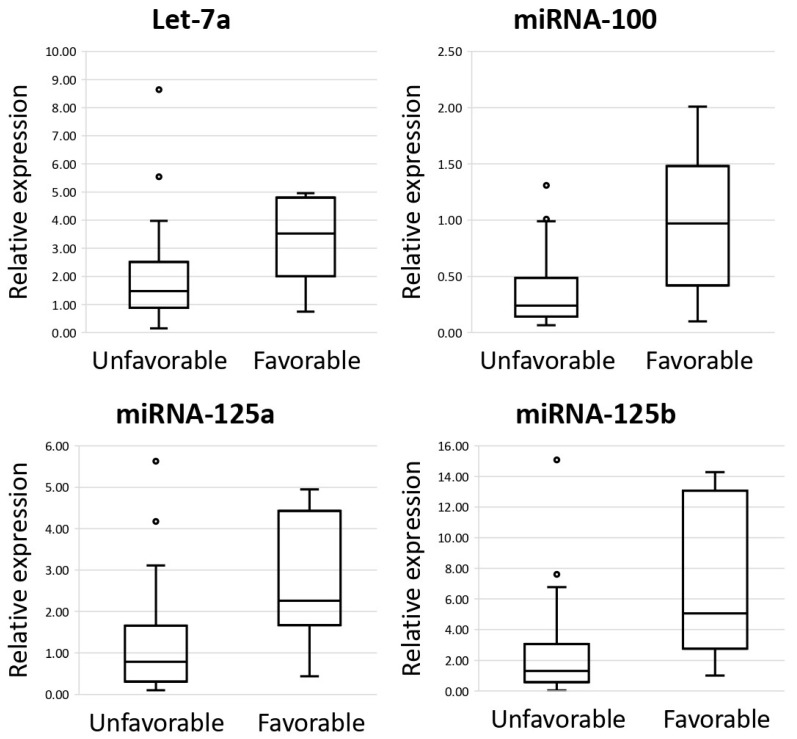
The comparative analysis of miRNA expression levels between diffuse large B-cell lymphoma (DLBCL) groups with an unfavorable (*n* = 36) and favorable (*n* = 7) prognosis. The figure presents medians, upper and lower quartiles, nonoutlier ranges, and outliers appearing as circles.

**Figure 4 cancers-17-01300-f004:**
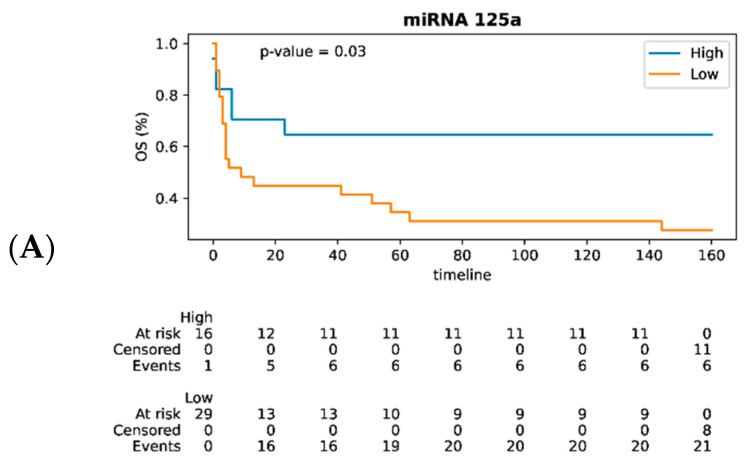
Analysis of the overall survival in DLBCL patients. (**A**) miRNA-125a; (**B**) miRNA-100; and (**C**) miRNA-23a. The blue and yellow curves denote patients with high and low miRNA expression levels, respectively.

**Figure 5 cancers-17-01300-f005:**
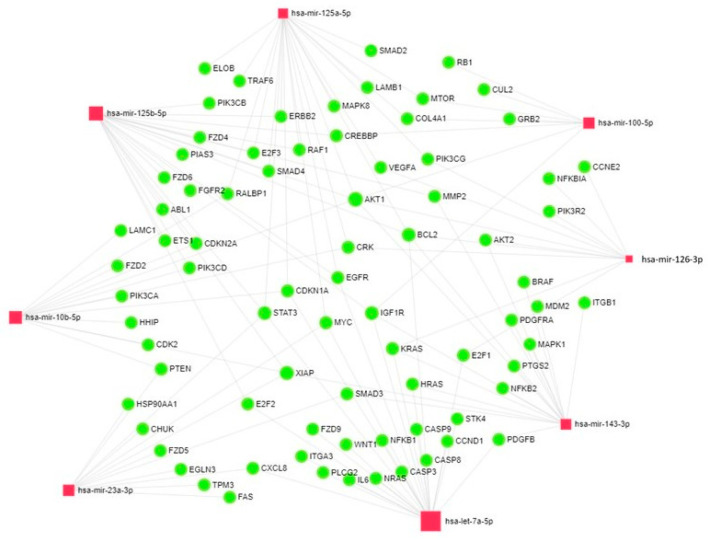
Targeted analysis of miRNA-10b, -100, -125a, -125b, -126, -143, -23a and let-7a using the miRnet 2.0 platform. Red and green rhombi denote microRNAs and their target genes, respectively.

**Table 1 cancers-17-01300-t001:** Comparative analysis of genes’ expression levels between bone marrow samples from patients with DLBCL and non-cancerous blood diseases.

	DLBCL Without Bone Marrow Involvement		DLBCL with Bone Marrow Involvement	
	Fold Change	Adjusted *p*-Value	Fold Change	Adjusted *p*-Value
*ASF1B*	1.18	NS	2.21	4 × 10^−4^
*CD82B*	−1.05	NS	1.60	NS
*CRISP3*	1.82	2 × 10^−3^	3.25	NS
*FN1*	2.67	1 × 10^−2^	1.92	NS
*MEF2B*	1.43	NS	1.67	NS
*PD-L1*	3.29	4.5 × 10^−5^	3.90	5 × 10^−5^
*TIMP1*	−2.48	5.05 × 10^−9^	−2.58	4.9 × 10^−4^
*TOP2A*	−2.62	6.8 × 10^−5^	−2.58	6.2 × 10^−4^
*TP53*	−6.55	1.5 × 10^−5^	−6.64	3 × 10^−5^

NS—not significant.

## Data Availability

All data generated or analyzed during this study are available from the corresponding author and will be provided on request.

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
