# Peer review of "MicroRNAs in Diffuse Large B-Cell Lymphoma (DLBCL): Biomarkers with Prognostic Potential"

_cancers, 2025, doi:10.3390/cancers17081300_

Round 1

Reviewer 1 Report (Previous Reviewer 1)

Comments and Suggestions for Authors

The manuscript titled " MicroRNAs in diffuse large B-cell lymphoma (DLBCL): Biomarkers with clinical potential" attempts to explores the role of deregulated microRNAs in DLBCL, aiming to identify novel biomarkers for diagnosis and prognosis while enhancing personalized therapy and understanding disease pathogenesis. The following improvements are necessary:

  1. The authors should reconsider the title to better align with the concluding statement in the discussion, which states that no statistically significant differences were found in the expression of the studied miRNAs based on IPI status. The revised title should accurately reflect the study's findings and avoid implying a definitive association where none was observed.
  2. Please provide list of PCR primers utilized i the study.
  3. Authors need to provide better quality representation image for figure 1. its unclear i present version to interpret.
  4. In fig1, what clustering method and distance metric were used in the hierarchical cluster analysis, and how do they influence the interpretation of miRNA expression patterns between DLBCL and NCBD cases? Also provide statistical tests performed to assess the robustness and significance of the observed clustering patterns, and how was variability within each group accounted for?
  5. Please mention how qPCR data was normalized in figure2 and 3.
  6. Have authors utilized any other method than Cox regression analyses for survival rate assessment?
  7. The discussion section could be more concise, as some details appear unnecessary or overly elaborate. Additionally, refining the interpretation of results would help ensure a clearer and more balanced analysis.

Author Response

Reviewer 2 Report (Previous Reviewer 2)

Comments and Suggestions for Authors

The author has revised the manuscript, and the updated version addresses my previous questions. The new version clearly distinguishes between samples from bone marrow and lymph nodes, making the study much easier to follow.

Minor comments:

  • In Supplementary Table 1, the "Response to Therapy" column appears to be empty. Could the author confirm whether this is intentional?
  • Figure 1 presents a heatmap of differentially expressed miRNA levels per sample. Please include a color key for expression levels.

Author Response

Reviewer 3 Report (Previous Reviewer 3)

Comments and Suggestions for Authors

The study is well conducted, sophisticated and brings an interesting point of view in the field of aggressive lymphomas.
Although it uses an investigation path already explored in other types of tumors, it suggests an original possibility to be integrated into future research, to differentiate and characterize diffuse large cell lymphomas in a more biologically complete way.
I have no particular criticisms to make of the work in this further submission, but there remains the practical difficulty of using such possible markers (miRNA levels) in routine and the difficulty in repeating the findings, to be used and confirmed in further research.
A future suggestion is to invest in a correlation of such sophisticated analyses, with parameters that are easier to compare. For example, radiological images of patients (CT and PET) with particular types of responsive or refractory DLBCL, with particular miRNAs, to be correlated with radiomics software. So that we can have a link that is easier to use in routine (Images), as a marker that still uses an underlying biological correlation, to better define the prognosis.

Round 2

Reviewer 1 Report (Previous Reviewer 1)

Comments and Suggestions for Authors

In the updated manuscript “MicroRNAs in diffuse large B-cell lymphoma (DLBCL): Biomarkers with clinical potential” the authors have successfully addressed all previous concerns. The manuscript now convincingly provides valuable insights into their potential of miRNA-23a, miRNA-125a, and miRNA-100 were shown to be potential prognostically significant biomarkers in DLBCL patients. Therefore, I recommend this article for publication.

This manuscript is a resubmission of an earlier submission. The following is a list of the peer review reports and author responses from that submission.

Round 1

Reviewer 1 Report

Comments and Suggestions for Authors

In the manuscript titled " MicroRNAs in diffuse large B-cell lymphoma (DLBCL): Biomarkers with clinical potential" claims that miRNA-23a, miRNA-125a, and miRNA-100 were prognostically significant biomarkers in DLBCL patients. While the manuscript is interesting, however it requires a following improvement to enhance clarity and impact:

1. The authors should provide IPI scores for the human patient samples. This scoring system, which considers factors such as age, performance status, disease stage, LDH levels, and extranodal involvement, is crucial for stratifying patients into low-, intermediate-, and high-risk groups. Including this data would offer a meaningful contrast to the miRNA profiling and underscore the prognostic value of the biomarkers identified in DLBCL patients.

2. The data summarized in Tables 1 and 2 should be supplemented with the original graphical representations used to derive the tables. Displaying the data visually through graphs will not only improve the manuscript's readability but also enable a clearer interpretation of the results.

3. The source of patient samples used for mRNA assessment should be explicitly detailed in the legend or results section accompanying Figure 1. 

Reviewer 2 Report

Comments and Suggestions for Authors

Diffuse Large B Cell Lymphoma (DLBCL) is a highly heterogeneous hematologic malignancy. This manuscript aims to enhance diagnostic approaches by profiling miRNAs in DLBCL patients. The authors isolate RNA from paraffin-embedded tissue sections and cytologic specimens, followed by RNA sequencing and miRNA-mRNA interaction analysis. In their study, the authors identify the upregulation of three miRNAs, which are significantly associated with poor prognosis in DLBCL patients.

Comments:

1.                  It would be beneficial to include information about the therapies administered (e.g., radiation, chemotherapy) and the therapy response (e.g., relapse, complete remission). Given that the manuscript compares miRNA profiles based on prognosis, this additional information could aid readers in better understanding and analyzing the data in the context of treatment outcomes.

2.            The authors analyze miRNAs from lymph node (LN) and bone marrow (BM) specimens separately. To enhance clarity, could the authors consider revising the subtitles for sections 3.1 and 3.2? This will help readers clearly understand that these sections pertain to miRNA analysis from DLBCL LN specimens specifically.

3.                  In the analysis, 219 miRNAs are differentially expressed in LN samples from patients with good versus poor prognosis. Could the authors provide a supplementary table summarizing these miRNAs, including the Log2 fold change, p-value, and adjusted p-value? Additionally, when selecting miRNAs for qPCR validation, what criteria were used to choose specific candidates? The manuscript highlights down-regulated miRNAs in patients with poor prognosis, but were any up-regulated miRNAs identified in poor prognosis specimens?

4.                  Sample Size for RT-PCR Validation (Line 198):
The authors report that the poor prognosis group consists of 36 patients and the favorable prognosis group consists of 8 patients. However, the LN specimen count is 43. Could the authors clarify why one additional sample was included for RT-PCR validation?

5.                     Line 189, The authors mention that a p-value of <0.01 was considered statistically significant. However, the manuscript employs a p-value threshold of <0.05. Could the authors verify and reconcile this discrepancy?

6.                     Line 212, It appears that there may be a typographical error in labeling the curve color in the manuscript. Specifically, the yellow curve is likely intended to be the red curve. Could the authors verify this and correct it if necessary?

7.                     Did the authors assess the mRNA levels of the target genes of the miRNAs identified in the study? For example, is there a upregulation of BCL-2 mRNA levels in LN samples from poor prognosis specimen?

8.                     Line 245, The manuscript reports that there are 18 patients with bone marrow involvement and 52 without. However, according to Supplementary Table 1, the total number of bone marrow specimens is listed as 60. Could the authors clarify this discrepancy? Moreover, how do the authors define bone marrow involvement? Is it based on histological analysis?

9.                  In the comparison of miRNA expression between poor and good prognosis LN samples, 219 miRNAs show differential expression. Additionally, 44 miRNAs are differentially expressed in DLBCL versus non-DLBCL bone marrow samples. Are there any overlapping miRNAs between these two datasets?

10.              The manuscript reports that DLBCL patients with bone marrow involvement show high expression of miR-92a and miR-126. Are these two miRNAs associated with poor prognosis, and if so, could the authors discuss their potential role in DLBCL progression?

11.              For consistency, could the authors standardize the formatting of miRNAs throughout the manuscript? Specifically, should it be written as microRNA-, miRNA-, or miR-?

Reviewer 3 Report

Comments and Suggestions for Authors

The study is very elegantly done and is very sophisticated in the description and identification of miRNAs as risk markers in the prognosis of DLBCL.

Also having linked the miRNAs involved in the prognosis of DLBCL with the main cellular pathways (apoptosis, BCR receptor pathway etc.) and with other types of cancer, to highlight the similar pathogenetic mechanisms in several different types of cancers, is a very interesting choice and offers further innovative research ideas to find points of union between different pathologies and potential ideas for drugs that can be functional between different pathologies.

However, the work is essentially descriptive and does not, for example, relate the aforementioned miRNAs with the 6 known genetic categories of DLBCL in order to allow further definitions of the risk, in relation to what is known today. Furthermore, by not establishing such a relationship, it remains a description among many, of the risk factors of DLBCL, which however, having neither a simple reproducibility in the clinical routine, nor a therapeutic implication, are not of high practical value, but only serve as a basis for further research. This somewhat reduces the value of the paper, which nevertheless remains of valuable development and may be useful for future research, which has a greater push towards direct utility on patients and on the clinic.
